# Complex Management of Nephrotic Syndrome and Kidney Failure during Pregnancy in a Type 1 Diabetes Patient: A Challenging Case

**DOI:** 10.3390/jcm11195725

**Published:** 2022-09-27

**Authors:** Leo Drapeau, Mathilde Beaumier, Julie Esbelin, François Comoz, Lucile Figueres, Giorgina Barbara Piccoli, Delphine Kervella

**Affiliations:** 1Néphrologie et Immunologie Clinique, CHU de Nantes, Nantes Université, 44000 Nantes, France; 2Néphrologie, Centre Hospitalier Public du Cotentin, 50100 Cherbourg, France; 3Service de Gynécologie-Obstétrique, CHU de Nantes, 44000 Nantes, France; 4Anatomie et Cytologie Pathologiques, CHU Caen Normandie, 14033 Caen, France; 5Néphrologie, Centre Hospitalier Le Mans, 72037 Le Mans, France; 6Center for Research in Transplantation and Translational Immunology, UMR 1064, ITUN, Inserm, CHU de Nantes, Nantes Université, F-44000 Nantes, France

**Keywords:** pregnancy, type 1 diabetes, nephrotic syndrome, acute kidney injury

## Abstract

Pregnancy with chronic kidney disease is challenging, and patients with diabetic nephropathy are at particular risk of a rapid kidney function decline during pregnancy. While indications for the management of pregnant patients with initial diabetic nephropathy are widely available in the literature, data on patients with severe nephrotic syndrome and kidney function impairment are lacking, and the decision on whether and when dialysis should be initiated is not univocal. We report a type 1 diabetes patient who started pregnancy with a severe nephrotic syndrome and shifted from CKD stage 3b to stage 5 during pregnancy. The management was complicated by a fetal heart malformation and by poorly controlled diabetes. The evidence for and against starting dialysis was carefully evaluated, and the choice of strict nephrological and obstetrical monitoring, nutritional management, and diuretic treatment made it possible to avoid dialysis in pregnancy, after ruling out pre-eclampsia. This experience enables examination of some open issues and contributes to the discussion of when to start dialysis in pregnancy.

## 1. Introduction

Diabetic nephropathy is a relatively frequent kidney disease in young women, and its management in pregnancy has been extensively studied, in particular in its initial stages [1,2,3,4,5]. However, less is known about the management of severe diabetic nephropathy in pregnancy [6,7]. The prognosis has been considered so grim by some authors that, in 2012, a review in a reference journal suggested that patients with severe diabetic nephropathy with a glomerular filtration rate reduced to ≤30% or other severe microangiopathic diseases should be advised to avoid pregnancy, and termination of pregnancy might occasionally be considered to prevent severe fetal and maternal morbidity [8,9]. Indeed, similar to other kidney diseases, the rate of decline in the glomerular filtration rate was reported to be comparable in women with early diabetic nephropathy and normal kidney function with or without pregnancy, while the cut-point of a creatinine higher than 124 μmol/L was identified as a risk marker for kidney disease progression in earlier studies [10,11]. 

While diabetic nephropathy shares with all chronic kidney diseases an increased incidence of hypertensive disorders of pregnancy and of preterm delivery, some risks are more specific. On the mother’s side, this includes a higher risk of kidney disease progression in the presence of full diabetic nephropathy. On the offspring’s side, these include a higher risk of malformations, particularly related to poor maternal glycemic control in the preconception and early pregnancy periods, and of both intrauterine and neonatal death [12,13,14,15]. Fetal growth may be differently affected by diabetes. While macrosomia is a marker of suboptimal diabetes control, also associated with intrauterine or neonatal death, early or severe placental impairment may lead to small-for-gestational-age babies [14]. Thus, placental impairment based on initial macrosomia may lead to a “false normal” birth weight, which is the result of the intrauterine growth restriction on a previously macrosomic fetus. In Europe, the incidence of severe congenital malformations is reported to be present in ∼4–6% of children born to diabetic mothers, and the perinatal mortality rate is reported to be as high as ∼3%. Moreover, it is estimated that while intrauterine growth restriction, preeclampsia, and congenital malformations may explain half of these deaths, in about 50% of the cases of perinatal death, the cause remains unknown [16,17,18,19]. However, many of these studies are old, and the impact of nephropathy is unclear, although given the higher risk of hypertensive disorders of pregnancy in full diabetic nephropathy, they are probably further increased in patients with overt nephropathy.

Rapid progression of diabetic nephropathy during pregnancy, as this case discusses, may introduce the choice of whether to start hemodialysis or continue with “conservative” therapy.

## 2. The Case

A 29-year-old woman with a history of type 1 diabetes from 11 years of age, complicated by multiple ketoacidosis episodes and proliferative retinopathy, was referred to nephrology at the beginning of her second pregnancy (10 gestational weeks (GW)) with nephrotic syndrome (urine protein/creatinine ratio (UPCR) 7 g/g, serum albumin 20 g/L, hypertension, serum creatinine 1.6 mg/dL, and an estimated glomerular filtration rate (eGFR, CKD-EPI formula 43 mL/min/1.73 m^2^) (Figure 1). The first pregnancy (6 years previously) was complicated by hypertension with a urinary protein-to-creatinine ratio (UPCR) of 0.8 g/g at the end of her pregnancy, without fetal growth impairment, and with normal kidney function. 

In her second pregnancy, due to the kidney function impairment with nephrotic proteinuria from the start of the pregnancy, a kidney biopsy was performed at 11 GW. The biopsy showed diabetic glomerulosclerosis with extensive interstitial fibrosis (60–80%), 20% of sclerotic glomeruli, and negative immunofluorescence (Figure 2). Diabetes control was poor despite insulin pump therapy, and the glycated hemoglobin was 10.4%.

The subsequent weeks of gestation were marked by a rapid decline in the eGFR, persistent nephrotic syndrome, and hypertension, leading to treatment with a calcium channel blocker.

At 25 GW, fetal ultrasounds revealed a small fetus (first percentile) with normal fetal Doppler and with a congenital cardiac malformation (transposition of the great arteries). In-hospital monitoring was required from 29 GW because of uncontrolled hypertension with increasing edema and further reduction in the kidney function. Optimization of the insulin pump therapy and equilibration of the diet resulted in better glycemic control, even though the values of the glycated hemoglobin were unreliable due to the concomitant anemia (HbA1c 5.9% at 29 GW + 3, with a hemoglobin level of 8.5 g/dL). The ratio of the soluble fms-like tyrosine kinase 1 (sFlt-1) to the placental growth factor (PlGF) was normal over time (ratio 5.0 at 25 GW, 2.9 at 29 GW, and 5.6 at 31 GW). There was no biological evidence for thrombotic microangiopathy, hemolysis, elevated liver enzymes, or low platelets (HELLP) syndrome. Corticotherapy for fetal lung maturation induction was administered at 28 + 5 GW (betamethasone 12 mg per day for two days).

The question of whether to start dialysis was discussed within a multidisciplinary team. Considering that the main reason for starting dialysis would have been the reduction in volume overload in the presence of a relatively low urea level (20 mmol/L), low-dose diuretic treatment (furosemide from 60 to 160 mg per day) was started to control volume overload and a moderately protein-restricted plant-based diet (protein intake 0.8 g/kg real weight/day), corresponding to the suggested protein intake in a non-pregnant individual without CKD, was started. The blood pressure was controlled with calcium channel blockers (Nicardipine 20 mg, according to blood pressure) and labetalol (200 mg every 8 h). Prudent curative anticoagulation with calcium heparin for a target anti-Xa activity of 0.3 was administered, given the severe nephrotic syndrome. The serum creatinine stabilized around 4.4 mg/dL (eGFR 12 mL/min/1.73 m^2^), urea around 20 mmol/L (BUN 56.1 mg/dL), normal potassium level and bicarbonates around 20 mmol/L, which allowed continuing the pregnancy for 4 more weeks. Meanwhile, hydramnios progressively developed (amniotic fluid index 240 mm at 30 GW and 310 cm at 31 GW) with intrauterine growth restriction (5th percentile at 30 GW) and a cerebral–placental ratio close to 1 (reflecting the redistribution of cardiac output to the cerebral circulation). The pregnancy was continued as long as possible to allow fetal growth because of the cardiac malformation (the required surgery is usually only possible when the baby reaches a weight of 3 kg). Delivery was induced at 33 + 3 GW because of both fetal Doppler alterations and worsening hypertension, resulting in the birth of a boy weighing 1980 g (22nd percentile), who immediately underwent a Rashkind balloon atrial septostomy. At three months (when his weight reached 3 kg), he successfully underwent an arterial switch operation. 

After delivery, maternal urea, creatinine, and albumin increased after introduction of a low dose of angiotensin-converting enzyme inhibitor, whereas the UPCR remained stable around 12 g/g. Five months after delivery, the patient started hemodialysis and is being prepared for kidney–pancreas transplantation (eGFR 6.8 mL/min, serum albumin 33 g/L).

## 3. Discussion

The patient herein described had many of the problems and challenges posed by patients with pregestational diabetes in pregnancy. 

Already at the start of her second pregnancy, her kidney function was impaired (CKD stage 3b), and intense proteinuria was present; both these elements are well acknowledged to be associated both with the risk of CKD progression during pregnancy and with adverse pregnancy outcomes [20]. Furthermore, the patient, who wanted to continue her pregnancy, had not been strictly followed in the pregestational phase. Diabetes was poorly controlled, and the kidney function and proteinuria data available did not allow a differential diagnosis between progressive diabetic nephropathy, usually characterized by a relatively regular progressive course, and another superimposed kidney disease. Hence, a kidney biopsy was performed in early pregnancy (12th gestational week) whose results showed severe diabetic nephropathy, notably characterized by intense interstitial fibrosis and glomerular sclerosis (Figure 2). 

While the risks of complications, and in particular of bleeding, are reported as increased when a kidney biopsy is performed in pregnancy, recent series are more reassuring on the risks, at least in early gestation and in experienced hands [21,22]. Even if, in her case, the treatment was not changed by the results, the findings supported a very strict followup as, given the extensive kidney fibrosis, both proteinuria and kidney function were expected to progressively worsen during pregnancy. 

Given the high blood pressure, we regularly tested biomarkers of the angiogenic–antiangiogenic balance (s-flt1 and PlGF) [23]. Even if these markers are not formally validated in CKD patients, they are increasingly employed in monitoring high-risk pregnancies [23,24,25]. In our case, the normal values suggested that placental dysfunction was not playing a major role in sustaining the high levels of proteinuria and hypertension, thus supporting an expectant management, and leading to consideration of dialysis, as discussed below.

The fetus presented one of the severe complications of pregestational diabetes, i.e., a cardiac malformation, which required surgery after delivery [26]. Moreover, the pregnancy was characterized by a combination of impaired fetal growth and polyhydramnios. Impaired fetal growth has been described in patients with type 1 diabetes, although the usual complication linked to poorly controlled diabetes is fetal macrosomia. In our case, the baby also presented with a congenital heart disease, and the mother had hypertension and chronic kidney disease, all of which probably played a role in fetal growth impairment [27]. The polyhydramnios may also have been of multifactorial origin; while diabetes mellitus is one of the most common causes of polyhydramnios, elevated urea levels may also play a role [28].

The rapid progression of renal function impairment led us to consider dialysis. It is difficult to assess the benefit–risk ratio of dialysasis in this context. Indeed, a small number of reports of “protective” or “preemptive” hemodialysis have recently been published. Wang and coworkers recently reported on a patient who started hemodialysis at 28 GW (in the presence of nephrotic proteinuria with a creatinine level of 343 μmol/L and a urea level of 15.1 mmol/L) [29]. Pregnancy was continued to 34 weeks, dialysis was discontinued thereafter, and the patient remained dialysis-free at 1 year of followup. The authors reported poor tolerance to hemodialysis, without further specification. Furthermore, in a large series from Australia, the pregnancy results were significantly better in patients who started dialysis in pregnancy, as compared to those who were already dialysis-dependent at conception, presumably also due to the positive effect of the residual kidney function [30]. However, data on the level of kidney function at the start of dialysis were not available in this registry analysis. The late start of dialysis was associated with higher risks of adverse pregnancy outcomes in a series mainly from Mexico, a country where, although dialysis is reimbursed in pregnancy, patients often have a delayed start with full uremic syndrome, both due to a late referral and to the reluctance to start treatment [6,31]. Conversely, reports of carefully followed cases from different settings underline that the start of dialysis may be delayed in pregnancy using a combination of drug treatments and nutritional management [32,33,34]. The high heterogeneity of the results and of the policies, including twin pregnancies, with early and late dialysis starts makes generalizations impossible [35,36]. However, in the absence of comparative studies, the advantages are not always clear, and the potential, albeit unproved, benefit of lowering the urea level must be balanced against other potential problems and contraindications (Table 1). These cautious considerations are likewise supported by a recent review of the available evidence for defining the best dialysis protocol in pregnancy, which mainly highlights the high dispersion of the data and the heterogeneity of the approaches, impairing clear-cut conclusions [37].

In our patient, the metabolic parameters (creatinine around 4.4 mg/dL, eGFR 12 mL/min/1.73 m^2^, and urea around 20 mmol/L, corresponding to BUN 56.1 mg/dL) would not have been considered as indications for starting dialysis outside the context of pregnancy. However, the urea was higher than the cutoff level associated with the worst outcome in the pivotal study by Hladunewich and coworkers in patients on chronic hemodialysis before pregnancy [38]. Nonetheless, no threshold has been established for the start of dialysis during pregnancy [39,40]. In our case, the urea level was stabilized under a diet with controlled protein intake (0.8 g/kg/day), in keeping with other reports at similar kidney function levels (none of which, however, were in patients with such an advanced stage of diabetic nephropathy) [41].

A further potential indication to start dialysis was the presence of severe edema, in the context of low albumin levels. The use of diuretics in pregnancy is generically contraindicated, due to the risk of placental hypoperfusion and of fetal growth impairment [42]. In the past, overzealous use of diuretics (especially intravenously) in preeclampsia was associated with worse pregnancy outcomes, and some cases of fetal ototoxicity were also reported [43]. However, loop diuretics are not associated with a teratogenic risk, and the need to continue or start diuretic treatment is well acknowledged in diabetic nephropathy [8]. Furthermore, the adverse effects of diuretic use were mainly due to the fact that preeclampsia, at least in its more severe forms, is often associated with relative hypovolemia, a condition that is not common in advanced CKD, although it may be encountered in nephrotic syndrome. Furthermore, several recent studies aiming to determine the safety of furosemide in pregnancy concordantly found that, at appropriate doses and under clinical control, these drugs can be safely used in pregnancy [44,45]. Volemic changes induced by ultrafiltration during hemodialysis, especially if associated with the low oncotic pressure in this setting of profound nephrotic syndrome, could likewise be detrimental to fetal–placental perfusion. In this regard, we felt that if the patient responded to diuretics, the volume depletion induced by this treatment was more gradual, with a lower risk of placental hypoperfusion. The polyhydramnios was relative stable during in-hospital followup. Furosemide therapy (administered sequentially) did not decrease the amniotic fluid volume. Overall, low furosemide doses, stabilization of urea, and better glycemic control may have participated in the stabilization of the hydramnios.

While the studies by Hladunewich and coworkers, reporting previously unrecorded good outcomes in pregnancy, reassure the feasibility of intensive hemodialysis, vascular access remains a problem [46,47]. In particular, in our case, with the presence of profound nephrotic syndrome, the risks of catheter infection and thrombosis were considered high. The vascular patrimony of type 1 diabetic patients is usually poor, and the creation of an arteriovenous fistula was not expected to be easy, necessary to enable a readily usable vascular access.

In our patient, the risk of exposing the patient and her baby to severe uremic toxicity was considered reasonably low, due to the stabilization of the kidney function with nutritional management. We also felt that, although the urea levels were higher than the target suggested for chronic dialysis patients, the conservative policy limited the risk of the rapid loss of residual kidney function after starting dialysis. Furthermore, the response to the diuretics was good, and the blood pressure was well controlled. In this setting, we felt that the potential advantages of delaying hemodialysis were offset by the risks, and we continued the expectant management until the presence of Doppler flow impairment and the restriction of fetal growth prompted delivery at 33 gestational weeks and 3 days. 

It is difficult to draw a clear conclusion from our case, which suggests that a close and concerted nephrological and obstetrical monitoring, including nutritional management, as an alternative to an early dialysis policy, may allow continuing pregnancy up to an acceptable term, with good fetal growth, even in complex and life-threatening situations (severe fetal malformation).

It is possible that an earlier dialysis start may have improved the prognosis, mainly retarding the need for delivery. However, it is impossible to say without randomized trials (which would be unethical in such a delicate context) or large observational studies comparing different policies. The decision should be shared with the patient, considering the experience and expertise of the different multidisciplinary teams, and carefully balancing what we know and do not about the early start of hemodialysis in pregnancy (Table 1).

## Figures and Tables

**Figure 1 jcm-11-05725-f001:**
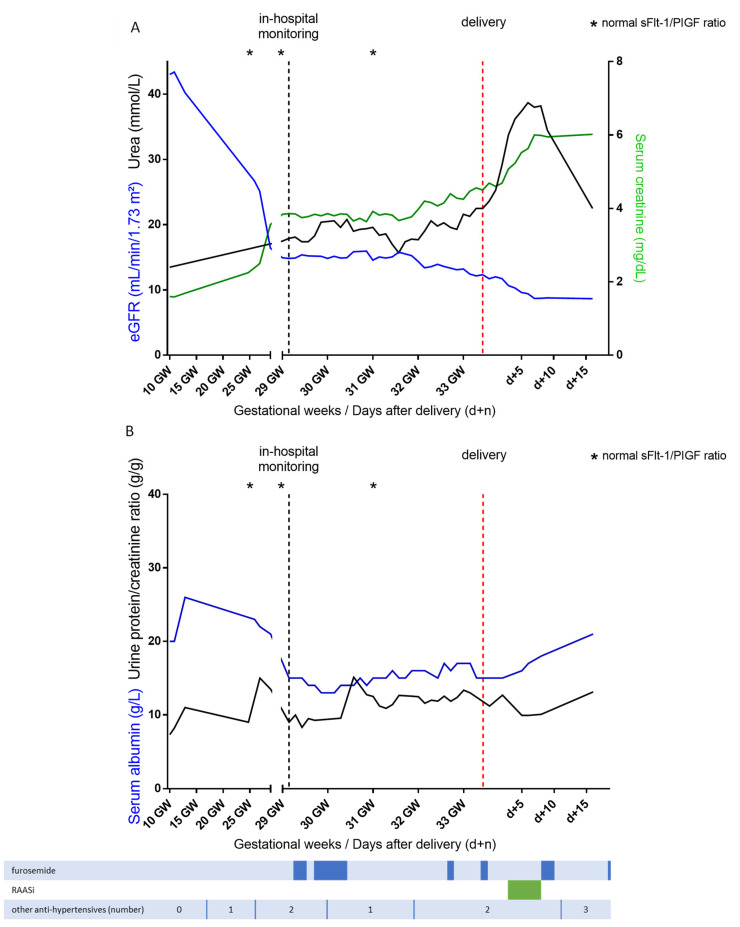
Evolution of the kidney function (**A**) assessed by serum creatinine (green line), estimated glomerular filtration rate (eGFR, blue line) and urea (black line) and nephrotic syndrome (**B**) assessed by urine protein/creatinine ratio (black line) and serum albumin (blue line) during pregnancy and after delivery. GW gestational weeks, RAASi Renin–angiotensin–aldosterone system inhibitor, sFlt-1/PIGF soluble fms-like tyrosine kinase 1-to-placental growth factor ratio.

**Figure 2 jcm-11-05725-f002:**
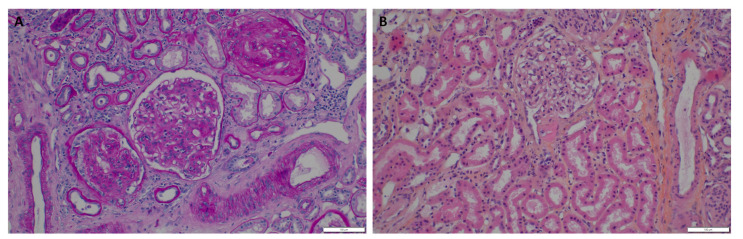
Renal histology (kidney biopsy performed at 11 gestational weeks). (**A**) Periodic acid Schiff staining. (**B**) Hematoxylin and eosin staining.

**Table 1 jcm-11-05725-t001:** Pros and cons of early dialysis initiation in pregnancy.

	Pros	Cons
Volume overload	Avoid loop diuretic use (risks of placental hypoperfusion and fetal growth impairment, ototoxicity)	Good effectiveness of loop diuretics on volume overload (slower depletion than during dialysis) Dialysis-induced volume shifts may be detrimental to placental perfusion
Metabolic disorders	Negative relationship between BUN level and birth weight	No threshold of BUN for dialysis start has been established during pregnancy
Other		Risk related to dialysis access placement (permanent central catheter) and permanence (infectious, thrombotic)

## Data Availability

The data presented in this study are available on request from the corresponding author. The data are not publicly available due to privacy.

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
