# Peer review of "Complex Management of Nephrotic Syndrome and Kidney Failure during Pregnancy in a Type 1 Diabetes Patient: A Challenging Case"

_jcm, 2022, doi:10.3390/jcm11195725_

Round 1
Reviewer 1 Report
Interesting case report on diabetic nephropathy and kidney failure during pregnancy.
Comments:
- When was the diabetes therapy switched to insulin pump (at or earlier than 29 weeks GA)? It would be interesting to plot the HbA1C levels on Figure 1.
- How do the authors explain the combination of impaired fetal growth with polyhydramnios? What was the evolution of the AFI (amniotic fluid index) during furosemide therapy? It would be interesting to plot EFW (estimated fetal weight) and the AFI, as obtained by ultrasound, in a figure.
Author Response
We would like to thank the editors and reviewers for reviewing our paper and providing interesting comments. Here we provide answers to the reviewers’ comments.
Interesting case report on diabetic nephropathy and kidney failure during pregnancy.
Comments:
- When was the diabetes therapy switched to insulin pump (at or earlier than 29 weeks GA)? It would be interesting to plot the HbA1C levels on Figure 1.
The patient was on insulin pump before pregnancy, however with poor glycemic control. Insulin pump therapy was optimized during in-hospital care. We added this information in the manuscript (line 83) “Diabetes control was poor despite insulin pump therapy, and glycated hemoglobin was 10.4 %”. Regarding HbA1c assessment, it was 5.9 % at 29GW+3, but at that time hemoglobin level was low (8.5 g/dL). Because of persistent anemia, we did not repeat HbA1c assessment. As HbA1c assessment are confounding because of anemia, we did not show the two values on the graph. We added this information in the manuscript (line 101): “Optimization of insulin pump therapy and equilibration of the diet resulted in better glycemic control even if the values of glycated hemoglobin were unreliable due to the concomitant anemia (HbA1c was 5.9 % at 29GW+3, with an hemoglobin level of 8.5 g/dL).”
- How do the authors explain the combination of impaired fetal growth with polyhydramnios? What was the evolution of the AFI (amniotic fluid index) during furosemide therapy? It would be interesting to plot EFW (estimated fetal weight) and the AFI, as obtained by ultrasound, in a figure.
Unfortunately, hydramnios assessment was described with different indexes (great cistern measurement before 30 GW, amniotic fluid index after 30 GW) so it is difficult to plot them together in a graph. We have added some information in the text:
Line 97: “At 25 GW fetal ultrasounds revealed a small fetus (first percentile)”
Line 121: “Meanwhile, major hydramnios progressively developed (amniotic fluid index 240 mm at 30 GW, 310 cm at 31 GW) with intrauterine growth restriction (5th percentile at 30 GW)…”
Since the patient had intricated pathologies, it is probable that both impaired fetal growth and polyhydramnios were multifactorial. Impaired fetal growth has been described in patients with type 1 diabetes, although the usual case is large-for-age babies. In our case, the addition of a congenital heart disease, hypertension and chronic kidney disease have probably favored the impairment of fetal growth. Polyhydramnios may also have been multifactorial with the implication of diabetes and elevated urea. Polyhydramnios was relatively mild and stable during in-hospital follow-up. Furosemide therapy (administered sequentially) did not decrease amniotic fluid volume. Overall, low furosemide doses, stabilization of urea and better glycaemic control may have participated in the stabilization of the hydramnios.
We have added these comments in the discussion (line 162 and line 225).
Reviewer 2 Report
The authors described an interesting case regarding the challenges of the optimal management of a diabetic pregnant patient with nephrotic syndrome and kidney failure. The manuscript is well written and the figures helpful. I ahve only few comments:
1. There was not mention about how the clinicians coped with the patient hypoalbuminemia. Was the patient under regular albumin infusions? Could hypoalbuminemia be considered as a risk factor of pre-eclampsia? What are the pros and cons of initiating dialysis in a pregnant patient with nephrotic syndrome in parallel with kidney failure?
2. In the discussion section the authors wrote in the lines 159-161 that "in the literature a small number of reports of “protective” or “preemptive” hemodialysis have recently been published...." Could the authors describe in more details the results of the other studies and compare the results and the differences of the other reported patients to the patient described in this case-report.
Author Response
We would like to thank the editors and reviewers for reviewing our paper and providing interesting comments. Here we provide answers to the reviewers’ comments.
The authors described an interesting case regarding the challenges of the optimal management of a diabetic pregnant patient with nephrotic syndrome and kidney failure. The manuscript is well written and the figures helpful. I have only few comments:
1.There was not mention about how the clinicians coped with the patient hypoalbuminemia. Was the patient under regular albumin infusions? Could hypoalbuminemia be considered as a risk factor of pre-eclampsia? What are the pros and cons of initiating dialysis in a pregnant patient with nephrotic syndrome in parallel with kidney failure?
We did not infuse albumin. To our knowledge there is no consistent data for the efficacy of albumin infusions (with or without furosemide) to reduce edema or other outcomes in nephrotic syndrome (Jacqueline J Ho et al., Cochrane Database Syst Rev 2019).
To our knowledge, hypoalbuminemia itself has not been evaluated as a risk factor of pre-eclampsia. Some papers suggest that preeclampsia is associated with lower albumin levels that may be associated with the severity of pre-eclampsia. In our case, there was no signs of preeclampsia.
The presence of a nephrotic syndrome may increase the risk of rapid volume shifts and low placental perfusion because of low oncotic pressure. Also, nephrotic syndrome increases the risk of infection and thrombo-embolic events, risks to consider in the case of central venous catheter placement.
We have added this in the discussion:
Line 221: “Volemic changes induced by ultrafiltration during hemodialysis, especially if associated to the low oncotic pressure in the setting of profound nephrotic syndrome, could likewise be detrimental to fetal-placental perfusion”.
Line 231: “In particular, in our case the presence of profound nephrotic syndrome, the risks of catheter infection and thrombosis were considered high”.
- In the discussion section the authors wrote in the lines 159-161 that "in the literature a small number of reports of “protective” or “preemptive” hemodialysis have recently been published...." Could the authors describe in more details the results of the other studies and compare the results and the differences of the other reported patients to the patient described in this case-report.
We have added a more detailed discussion of the other studies describing patients with worsening kidney function during pregnancy (lines 172-195). As described, there are no comparative studies in this setting and it is difficult to draw conclusions from mostly case reports.